# Quercetin: A Functional Food-Flavonoid Incredibly Attenuates Emerging and Re-Emerging Viral Infections through Immunomodulatory Actions

**DOI:** 10.3390/molecules28030938

**Published:** 2023-01-17

**Authors:** Fauzia Mahanaz Shorobi, Fatema Yasmin Nisa, Srabonti Saha, Muhammad Abid Hasan Chowdhury, Mayuna Srisuphanunt, Kazi Helal Hossain, Md. Atiar Rahman

**Affiliations:** 1Department of Environmental and Biological Chemistry, Chungbuk National University, Cheongju 28644, Republic of Korea; 2School of Allied Health Sciences, Walailak University, Nakhon Si Thammarat 80160, Thailand; 3Department of Biochemistry and Molecular Biology, University of Chittagong, Chittagong 4331, Bangladesh; 4Bangladesh Tea Research Institute, Sreemangal-3210, Moulvibazar District, Sylhet 3100, Bangladesh; 5Department of Neuroscience, Lerner Research Institute, Cleveland Clinic, 9620 Carnegie Avenue, Cleveland, OH 44195, USA

**Keywords:** quercetin, antiviral action, flavonoid, SARS-CoV-2, Dengue, Ebola, influenza, hepatitis C virus, mechanisms, medicinal plant

## Abstract

Many of the medicinally active molecules in the flavonoid class of phytochemicals are being researched for their potential antiviral activity against various DNA and RNA viruses. Quercetin is a flavonoid that can be found in a variety of foods, including fruits and vegetables. It has been reported to be effective against a variety of viruses. This review, therefore, deciphered the mechanistic of how Quercetin works against some of the deadliest viruses, such as influenza A, Hepatitis C, Dengue type 2 and Ebola virus, which cause frequent outbreaks worldwide and result in significant morbidity and mortality in humans through epidemics or pandemics. All those have an alarming impact on both human health and the global and national economies. The review extended computing the Quercetin-contained natural recourse and its modes of action in different experimental approaches leading to antiviral actions. The gap in effective treatment emphasizes the necessity of a search for new effective antiviral compounds. Quercetin shows potential antiviral activity and inhibits it by targeting viral infections at multiple stages. The suppression of viral neuraminidase, proteases and DNA/RNA polymerases and the alteration of many viral proteins as well as their immunomodulation are the main molecular mechanisms of Quercetin’s antiviral activities. Nonetheless, the huge potential of Quercetin and its extensive use is inadequately approached as a therapeutic for emerging and re-emerging viral infections. Therefore, this review enumerated the food-functioned Quercetin source, the modes of action of Quercetin for antiviral effects and made insights on the mechanism-based antiviral action of Quercetin.

## 1. Introduction

Viral infection has now become the biggest global concern for healthcare professionals because of the increasing incidence of morbidity and mortality. Viral infectious illnesses are becoming a severe hazard to human health in both developing and developed countries [1], killing 1 million people each year. Human immunodeficiency virus (HIV), hepatitis virus subtypes A, B, and C (HAV, HBV, and HCV), herpes simplex virus (HSV), influenza virus, Dengue virus, Ebola virus, and other viruses such as monkeypox/mpox virus have been affecting human health for decades [2]. Along with these pre-existing viruses, corona virus-2 (SARS-CoV-2) became a global burden beginning in 2019. Coronavirus infection, commonly known as “new coronavirus disease” (COVID-19), is characterized by severe acute respiratory illness with a significant mortality rate [3]. Unfortunately, many of the viral diseases are not yet curable, such as corona virus-2 (SARS-CoV-2), human immunodeficiency virus (HIV), hepatitis virus, human papillomavirus, and others. The emergence of new diseases, such as the Zika virus, Chikungunya virus, and Dengue virus, is creating new health challenges and significant concerns in healthcare.

The pathogenesis of viral infections has also been reported to provoke various types of complications that target different organs and the immune system and sometimes can even promote tumor progression. Due to the lack of safe, as well as effective, antiviral drugs against these viruses, the substantial health burden with direct and indirect costs, including hospitalization and loss of productivity [4], is increasing. Additionally, the toxicity and ineffectiveness of synthetic antiviral drug responses to resistant strains have fueled the hunt for effective and alternative treatment options, such as plant-derived antiviral medicinal molecules. Furthermore, vaccines and antiviral drugs are frequently and prohibitively expensive, making them inaccessible to most of the world’s population. As a result, the current review, as part of an effort to find natural treatments for viral infections, attempts to investigate the possibility, probable source, and mechanistic insights of a plant-based natural flavonoid Quercetin against SARS-CoV-2, HIV, and HBV, Dengue and Ebola virus.

Plant-based natural molecules are producing huge interest for the stakeholders, including the researchers, due to their effectiveness, less or no harmful effects, affordability and greater compliance of patience. Therefore, natural biometabolites are thought to be the best option for producing new medications. During the last few years, many researchers have been attempting to develop novel antiviral medications [5]. Natural compounds have grown in popularity in recent decades due to their diverse biological roles and direct application as treatments [5,6]. Quercetin is a distinct flavonoid component renowned for its antiviral effects. It has fewer or no negative effects than synthetic drugs and could be a viable therapeutic option for certain viral infections [6].

Quercetin (3,3′,4′,5,7-pentahydroxy-2-phenylchromen-4-one) (Figure 1), the major representative of the flavonoid subclass of flavonols [7], is derived from the Latin word “Quercetum,” meaning “Oak Forest” [8]. Quercetin, yellow in appearance, is availably found in a variety of vegetables and fruits, including lovage, capers, berries, cilantro, dill, apples, and onions [9]. Quercetin is entirely soluble in lipids and alcohol, slightly soluble in hot water but insoluble in cold water [8].

Quercetin is an active medicinal drug that has antiviral activity and suppresses viruses by targeting viral infections at various stages. Several studies have emphasized the potential utility of Quercetin as an antiviral due to its capacity to suppress the early phases of virus infection, interact with viral replication proteases, and diminish infection-induced inflammation [10]. Quercetin may also be effective in combination with other medications to potentially boost their efficacy or interact synergistically with them to lessen their side effects and associated toxicity. The main molecular mechanisms of Quercetin’s antiviral effects are the inhibition of viral neuraminidase, proteases, DNA/RNA polymerases, and the modification of several viral proteins. Quercetin has been reported to have an antiviral effect against several viruses, including the hepatitis C virus (HCV) [11], the Mayaro virus [12], the influenza A virus (IAV) [13], the Chikungunya virus (CHIKV) [14,15]. It has also been documented to suppress HCV by binding to and inactivating the viral NS3 protease [16]. It is literally cited to inhibit viral RNA polymerase, preventing DENV-2 replication [17] and suppressing Ebola virus infection via VP24 Interferon-Inhibitory Function [18]. Our recent studies, using computer models, have shown the effectiveness of Quercetin against the Hepatitis C virus, Dengue type 2 virus, Ebola virus, and Influenza A. To discover the optimal binding sites between Quercetin and PubChem-based receptors, molecular docking research was conducted using the web program PockDrug. A strong binding affinity of Quercetin was demonstrated for the HCV NS5A protein (docking score of—6.268 kcal/mol), the DENV-2 NS5 protein (docking score of—5.393 kcal/mol), the EBOV VP35 protein (docking score of—4.524 kcal/mol) and the IAV NP protein (docking score of—6.954 kcal/mol). The Search Tool for the Retrieval of Interacting Genes/Proteins (STRING), the search tool for interactions of chemicals (STITCH), Gene Set Enrichment Analysis (GSEA,) and the Cytoscape plugin cytoHubba network-pharmacological tools were used to confirm function-specific, gene-compound interactions demonstrating the 38 genes that interacted with Quercetin. The top interconnected nodes in the protein–protein network was AKT1 (Serine/threonine protein kinase), Proto-oncogene tyrosine-protein kinase SRC (SRC), Epidermal growth factor receptor (EGFR), matrix-metalloproteinase (MMP9), kinase insert domain receptor (KDR), MMP2, insulin-like growth factor receptor 1 (IGF1R), protein tyrosine kinase 2 (PTK2), breast cancer resistance protein (ABCG2) and MET [19]. However, the differences in their mechanistic insights, use of Quercetin-rich plants for future antiviral drug discovery and immunomodulatory actions of Quercetin to defend against viral infections are yet to be fully enumerated. Therefore, this review discusses the mechanistic insights of Quercetin against influenza A, Hepatitis C, Dengue type-2, and Ebola virus; the direction of using the observed effects (Figure 2) for advanced antiviral drug discovery; and the association of antiviral actions with immunomodulatory performances to further suggest the therapeutic prospects of Quercetin/its derivatives as antiviral agent/food supplements.

## 2. Methodology and Resources

An extensive literature search was conducted to gather all relevant information. Publicly accessible databases and primary sources, including CNKI, PubMed, SciFinder, Web of Science, and ScienceDirect were searched. Many relevant published articles were critically reviewed. All figures were drawn with MS PowerPoint and drawing instruments. The keywords flavonoids, Quercetin, antiviral compounds, phytocompounds for antiviral action, and antiviral mechanism of natural compounds were chosen for the comprehensive search of materials.

## 3. Research Question and Hypothesis

Many plant-based flavonoids are used as antimicrobial, antioxidant, anti-inflammatory, anticancer, and antiaging agents. Several different mechanisms are also proposed for how those functions are achieved by Quercetin. Quercetin is also used as a reference agent in some of the in vitro radical scavenging activities eventually leading to showing antioxidant effects. Antioxidant agents have been reported to be effective as antiviral in some of the in vivo and cellular studies. Thus, the antiviral action of the well-known strong antioxidant agent Quercetin is inevitable to be resurgenced for its greater therapeutic benefit. Quercetin is a polyphenolic that belongs to the flavonol class of flavonoids found widely among vegetables and fruits and is a regular component of a normal diet, which is not found in the human body [8]. This biological function of Quercetin is thought to be contributed by the pentahydroxy flavone with five hydroxy groups at positions 3, 3’, 4’, 5, and 7 [20]. It is a Quercetin-7-olate conjugate acid. Glycosides and ethers are the most common groups of Quercetin derivatives, and sulfate and phenyl substituents are the less common [20]. Rutin (Quercetin-3rutinoside) is the most common source of Quercetin [21]. As a nutritional supplement, Quercetin is well-tolerated. Quercetin has shown many pharmacological activities, which include anti-inflammatory [22], antiproliferative [23], antioxidative [24], antibacterial [25], anticancer [26], neuroprotective, hepatoprotective [27,28], and antiviral [29,30,31] activities. Quercetin also increases mitochondrial biogenesis while inhibiting platelet aggregation, capillary permeability, and lipid peroxidation [8]. The current review aims to understand the mechanistic insights of four different types of viruses and the role of Quercetin in protecting against their infections; natural sources of Quercetin and its derivative; and the immunomodulatory contribution of Quercetin in controlling viral infections.

## 4. Natural Sources of Quercetin and Its Isolation from Plants

Quercetin is one of the most consumed and important bioflavonoid components and is widely found in different varieties of fruits and vegetables. Plant species, growing conditions, harvest conditions, and storage methods can influence the polyphenolic composition of fruits and vegetables. Quercetin is found in abundance in onions, apples, and wine. According to several studies, Quercetin is also found in tea, pepper, coriander, fennel, radish, and dill [32]. More than 20 plants species produce Quercetin: *Foeniculum vulgare*, *Curcuma domestica* valeton, *Santalum album*, *Cuscuta reflexa*, *Withania somnifera*, *Emblica officinalis*, *Mangifera indica*, *Daucus carota*, *Momordica charantia*, *Ocimum sanctum*, *Psoralea corylifolia*, *Swertia chirayita*, *Solanum nigrum*, and *Glycyrrhiza glabra*, *Morua alba*, *Camellia sinensis* [33], *Allium fistulosum*, *A. cepa*, *Calamus scipionum*, *Moringa oleifera*, *Centella asiatica*, *Hypericum hircinum*, *H. perforatum*, *Apium graveolens*, *Brassica oleracea* var. *Italica, B*. oleracea var. sabellica, *Coriandrum sativum*, *Lactuca sativa*, *Nasturtium officinale*, *Asparagus officinalis*, *Capparis spinosa*, *Prunus domestica*, *P. avium*, *Malus domestica*, *Vaccinium oxycoccus*, and *Solanum lycopersicum* [9]. Quercetin is available in capsule and powder form as a dietary supplement. The plasma Quercetin concentration rises when Quercetin is consumed in the form of foods or supplements (Table 1). As a result, everyday consumption of Quercetin-rich foods increases Quercetin bioavailability and contributes to the prevention of lifestyle-related disorders [32].

Quercetin was isolated from a fractionated extract of *Rubus fruticosus* by using an optimized column in HPLC and increasing its concentration by using a nanofiltration membrane [34]. Extraction of Quercetin from different plant sources can be followed by effective sample preparation techniques known as the sea sand disruption method (SSDM). The SSDM is used due to its recovery efficiency [35]. During the isolation of Quercetin and its derivatives in plants’ source, SSDM is used to eliminate errors in the study [36]. Flavonoids are isolated from the crude extract of plants by using various organic solutions followed by HPLC analysis, which is further characterized by FTIR, NMR, and mass spectroscopy [37]. Quercetin-3-*O*-rhamnoside was isolated from *P. thonningii* leaves by using different organic solvents [38]. According to one study, dihydroQuercetin, one of the Quercetin derivates, was isolated from Larix gmelinii using ultrasound-assisted and microwave-assisted alternate digestion methods because they required less extraction time, less energy, and were more cost-effective than conventional solvent extraction methods [39]. Another derivative known as Isorhamnetin was isolated from the crude extract of *Stigma maydis* through two-stage high-speed countercurrent chromatography processes, where two-phase solvent systems composed of n-hexane-ethyl acetate-methanol-water are used at volume ratios of 5:5:5:5 and 5:5:6:4 to ensure the purity [40].

**Table 1 molecules-28-00938-t001:** Quercetin and its derivatives from different plant sources and their biological effects in various experimental models.

Phytochemical	Plant Name	Family	Plant Parts	Virus Target	Cell	Bioassay	Viral Step or MOA	Reference
Quercetin-3-o-α-L-rhamnopyranoside (Q3R)	*Rapanea melanophloeos*	Myrsinaceae	Whole plant	IAV	MDCK cell	In vitro	Inhibit viral entry and virus replication	[41]
Quercetin 3-glucoside	*Dianthus superbus* L.	Caryophyllaceae	Whole plant	IAV	MDCK cell	In vitro and in silico	Inhibit viral replication	[42]
Quercitrin (Quercetin-3-L-rhamnoside)	*Houttuynia cordata Thunb.*	Saururaceae	Leaf (Aerial parts)	IAV (Anti-influenza A/WS/33 virus)	Mammalian kidney (BHK)	In vitro	Inhibit replication in the initial stage of virus infection by indirect interaction with virus particles	[43]
Rutin (Quercetin-3-rutinoside)	*Prunus domestica*	Rosaceae	Fruit	HCV	Human hepatocellular carcinoma cells Huh 7 and Huh 7.5	In vitro and ex vivo	Inhibit the early stage of viral entry	[44]
Quercetin	*Psidium guajava*	Myrtaceae	Bark	DENV	Epithelial VERO cells (Cercopithecus aethiops)	In vitro and in silico	Directly inhibit the viral NS3 protein and could interrupt virus entry by inhibiting fusion	[45]
Quercetin	*Embelia ribes*	Myrsinaceae	Seeds	HCV	Huh-7 cells	In vitro	Inhibit NS3 protease activity and HCV replication.	[16]
Quercetin 7-rhamnoside	*Houttuynia cordata*	Saururaceae	Aerial Parts	Porcine epidemic diarrhea virus (PEDV CV 777)	Vero (african green monkey kidney cell line; ATCC CCR-81) ST (pig testis cell line; ATCC CRL-1746)	In vitro and In vivo	Inhibit at an early stage of viral replication after infection	[29]
Quercetin and its glycoside derivatives	*Bauhinia longifolia* (Bong.)	Fabaceae	Leaves	Mayaro viruses (ATCC VR-66, lineage TR 4675)	Vero cells (African green monkey kidney, ATCC CCL-81)	In vitro	glycosilation re duces the antiviral activity of Quercetin against	[12]
DihydroQuercetin (DHQ)	*Larix sibirica* (larch wood)	Pinaceae	Wood	Coxsackie virus B4 Powers strain	Vero cells Inbred, female mice	In vivo	Decrease the replication of viral protein by reducing ROS generation	[46]
Quercetin-7-o-glucoside	*Dianthus superbus*	Caryophyllaceae	Leaves	Influenz viruses A/Vic/3/75 (H3N2, VR-822), A/PR/8/34 (H1N1, VR-1469), B/Maryland/1/59 (VR-296) and B/Lee/40 (VR-1535D)	Madin-Darby Canine Kidney (MDCK) cell	In Vitro	Inhibit influenza viral RNA polymerase PB2	[24]
Quercetin and Isoquercitrin	*Houttuynia cordata*	Saururaceae	Whole plant	Herpes simplex virus (HSV)	African green monkey kidney cells (Vero, ATCC CCL-81) and human epithelial carcinoma cells	In vitro	Quercetin and isoquercitrin inhibit NF-κB activation in HSV viral replication	[47]
Kaempferol	*Rhodioila rosea*	Crassulaceae	Roots	The influenza strains A/PR/8/34 (H1N1) (ATCC VR-1469)	Madin-Darby canine kidney (MDCK) cells were obtained	In vitro	Inhibit viral replication by blocking neuraminidases	[48]
Myricetin	*Marcetia taxifolia*	Melastomataceae	Aerial parts	HIV-1 (HTLV-IIIB/H9)	MT4 cells	In silico	May Bind to NNRTI pocket of NNRTI resistant HIV-1	[49]
Apigein	*Gentiana veitchiorum*	Gentianaceae	Flower	Foot-and-mouth disease virus (FMDV)	BHK-21 cells	In vitro	Block the internal ribosome entry site (IRES) mediate translational activity	[50,51]
Quercetin 3-o-β-glucopyranoside	*Morus Alba*	Moraceae	Leaf	*Herpes simplex* Virus type 1	Vero cell line no ATCC CCL-81)	In vitro	Inhibit DNA chair termination	[52]
Quercetin 3-o-β-(6”-o-galloyl)-glucopyranoside	*Morus Alba*	Moraceae	Leaf	*Herpes simplex* Virus type 1	Vero cell line no ATCC CCL-81)	In vitro	Inhibit DNA chair termination	[52]
Quercetin-3-o-β-L-rhamnopyranosyl	*Acacia albdai*	Fabaceae	Leaf	*Herpes simplex* Virus type 1	Vero cell line no ATCC CCL-81)	In vitro	Inhibit DNA chain termination	[52]
Quercetin-3-*O*-α-L-rhamnopyranoside	*Acacia albdai*	Fabaceae	Leaf	*Herpes simplex* Virus type 1	Vero cell line no ATCC CCL-81)	In vitro	Inhibit DNA chain termination	[52]
6-o-methoxy Quercetin-7-o-β-D-glucopyranoside	*Centaurea glomerata*	Asteraceae	Aerial parts	*Herpes simplex* Virus type 1	Vero cell line no ATCC CCL-81)	In vitro	Inhibit DNA chain termination	[52]
4’,6-o-dimethoxy Quercetin-7-o-β-D-glucopyranoside	*Centaurea glomerata*	Asteraceae	Areal Parts	*Herpes simplex* Virus type 1	Vero cell line no ATCC CCL-81)	In vitro	Inhibit DNA chain termination	[52]
Quercetin-3-β-o-D-glucoside	*Allium cepa*	Amaryllidaceae	Root	Ebolaviruses (EBOV-Kikwit-GFP, EBOV Makona, SUDV-Boniface, mouse-adapted EBOV)	Vero E6 cells	In vitro	Block glycoprotein mediated step during viral entry	[53,54]
Isorhamnetin	*Ginkgo biloba*	Ginkgoaceae	Leaf	Influenza A virus Puerto Rico/8/34 (H1N1)	Madin Darby Canine Kidney (MDCK) cells	In vitro and In vivo	Inhibit neuraminidase and hemagglutination, suppress ROS generation and ERK phosphorylation	[55,56]
Luteolin	*Elsholtzia rugulosa*	Lamiaceae	Whole Plant	Influenza viruses A/PR/8/34(H1N1), A/Jinan/15/90(H3N2) and B/ Jiangsu/10/2003	MDCK cells	In vitro	Inhibit the neuraminidase	[57]
Luteolin	*Cynodon dactylon*	Poaceae	Whole Plant	Chikungunya virus	Vero cells	In vitro	Inhibit intracellular viral replication	[58]
Quercetin	*Illicium verum*	Schisandraceae		Singapore grouper iridovirus (SGIV)	Grouper spleen (GS) cells	In vitro	Interrupt SGIV binding to host cell by blocking membrane receptor on host cell which	[59]
Naringenin	*Citrus sinensis*	Rutaceae	Fruit	Zika Virus	Human A549 cells	In vitro	Inhibit NS2B-NS3 protease	[60,61]
Hesperidin	*Citrus sinensis* (sweet orange)	Rutaceae	Fruit Peel	SARS-CoV-2 virus		In silico	Binds to main protease and angiotensin converting enzyme 2	[62]
Hesperidin	*Citrus sinensis*	Rutaceae	Fruit Peel	Sindbis virus	BHK-2	In vitro	Inhibitory activity on viral replication	[63,64]
Naringenin	*Citrus paradisi*	Rutaceae	Fruit Peel	Hepatitis C virus (HCV)	Huh7.5.1 human hepatoma cell	In vitro and In vivo	inhibits ApoB lipoprotein reduce secretion of HCV	[65,66]
Luteolin	*Achyrocline satureioides*	Asteraceae	Whole Plant	Influenza virus A/Fort Monmouth/1/1947 (H1N1)	Madin-Darby canine kidney (MDCK) cells and Vero cells	In vitro	Block absorption to the cell surface or receptor binding site leads to the suppress of the expression of coat protein I	[67,68]
Naringenin	*Citrus paradisi*	Rutaceae	Fruit Peel	Dengue virus (DENV)	Huh7.5 cells	In vitro	Act as antiviral cytokine during DENV replication	[66,69]

## 5. Absorption, Metabolism, Distribution, and Excretion of Quercetin

Quercetin is taken as glycosides, with glycosyl groups released during chewing, digestion, and absorption. In humans, only a small percentage of Quercetin is absorbed in the stomach, and the primary site of absorption is the small intestine [70]. Two methods allow Quercetin glycosides to be absorbed in the intestine. One method is lactose polarizing hydrolase (LPH) in the brush border membrane, and another method is the interaction with the sodium-dependent glucose transporter (SGLT1) [33]. The gut microbiota plays a crucial role in the absorption of Quercetin by enzymatic hydrolysis. After absorption, the metabolism of Quercetin takes place in various organs, including the small intestine, colon, liver, and kidney. Biotransformation enzymes in the small intestine and liver create methylated, sulfated, and glucuronate forms of Quercetin metabolites due to phase II metabolism [32]. After that, these are released into the bloodstream via the portal vein of the liver. In the small intestine and colon, Quercetin metabolism leads to the generation of phenolic acids. The metabolites of Quercetin are found in human plasma as methylated glucuronide or unmethylated sulfate. The major metabolite of Quercetin, Quercetin-3-o-b-D-glucuronide, is delivered to target tissues via plasma to exert biological activity [32]. Quercetin had a short half-life and rapid clearance in the blood, and its metabolites appeared in the plasma 30 min after ingestion; however, considerable amounts were excreted over 24 h [71]. In comparison to other phytochemicals, Quercetin has a high bioavailability. The bioavailability of Quercetin decreases when consumed as a supplement rather than food. Quercetin is excreted from the human body in the feces and urine, and in high doses, it can be discharged through the lungs. 3-hydroxy phenylacetic acid, hippuric acid, and benzoic acid are the excretory products of Quercetin [32].

## 6. Major Pharmacological Actions of Quercetin

Flavonoids, particularly Quercetin, which has well-known antioxidant effects, are gaining popularity these days. Quercetin has been identified as a potential anticancer drug with activity both in in vitro and in vivo models. Quercetin is used to inhibit the spread of various cancers, such as lung, prostate, liver, breast, colon, and cervical cancers, by modifying oxidative stress factors and antioxidant enzymes [8]. Because of its chemoprotective action against tumor cell lines through metastasis and apoptosis, Quercetin is thought to be a promising anticancer option [72]. Furthermore, another study revealed the powerful efficiency of combined Quercetin-doxorubicin treatment in maintaining T-cell tumor-specific responses, resulting in better immune responses against breast tumor growth [73]. Antioxidants work against asthma pathogenesis by avoiding oxidative damage through a variety of methods. Quercetin plays a role in scavenging free radicals that can lead to cell death by damaging DNA and cell membranes. In addition, it has been noted that Quercetin decreases the production and release of histamine and other mediators involved in the development of allergic reactions in mast cells, suggesting that it could be effective against asthma [74]. In in vitro and in vivo studies, Quercetin has been shown to protect neurons from oxidative and neurotoxic chemicals, saving the central nervous system from oxidative stress-induced neurodegenerative diseases, especially Alzheimer’s disease (AD) and Parkinson’s disease (PD). Quercetin has been shown as anti-Alzheimer’s because it improves mitochondrial morphology, improves memory impairments, protects cognitive deficits, and reduces neurodegeneration [32].

## 7. Antiviral Actions of Quercetin

### 7.1. Quercetin against Hepatitis C Virus (HCV)

The hepatitis C virus (HCV) is a small, enveloped, positive-sense, single-stranded RNA virus of the family *Flaviviridae* [75] with seven major genotypes [76]. The hepatitis C virus is a bloodborne virus. An estimated 71 million people (1%) worldwide are infected with hepatitis C [77]. In the long run, this can lead to severe liver fibrosis, cirrhosis, and hepatocellular carcinoma. As a result, in developed countries, HCV is the most common reason for liver transplantation [78]. A new infection with HCV in some people clears spontaneously. If untreated, most patients with HCV become chronic carriers and are at risk for severe consequences, such as cirrhosis and hepatocellular carcinoma (HCC), and death [79]. The most effective secondary prevention of HCV-related chronic liver disorders is curative antivirals. Pegylated interferon with ribavirin (RBV) was once the standard HCV treatment. Unfortunately, 70–80% of HCV patients cannot receive interferon-based therapy because of severe effects or contraindications [80]. After that, Ledipasvir-sofosbuvir was a new interferon-free/RBV-free treatment option that helps patients avoid severe consequences [81,82]. However, the expense of this medication restricts availability, as a 12-week ledipasvir-sofosbuvir treatment costs $94,500 per patient, according to a cost-benefit study [83]. The current treatment method for HCV is pan-genotypic direct-acting antivirals (DAAs). DAAs can cure most patients with HCV infection, and treatment time is also short (usually 12 to 24 weeks), depending on the absence or presence of cirrhosis. Treatment with direct-acting antivirals (DAA) offers HCV cure rates of over 95% [84]. Access to HCV treatment is improving but remains too limited. The high price of oral DAAs is still a major barrier in low and middle-income countries. Since these drugs are not widely available in all parts of the world, there is a need to develop cost-effective strategies [85]. Quercetin, a bioflavonoid, has the potential to become a less expensive and more natural alternative to costly interferon-free/RBV-free therapy. A phase I clinical trial reported that chronically infected hepatitis C patients can take up to 5000 mg daily for 28 days without experiencing any side effects or aberrant laboratory results [86]. Due to its great tolerability, it may be safe for long-term usage in other affected populations, such as those with HCV-associated liver failure, and it could be used to prevent relapse.

### 7.2. Mechanism of Quercetin against HCV Virus

Quercetin, a flavonoid, has been proven to have anti-HCV effects through a variety of methods: It has been reported to reduce internal ribosomal entry site (IRES) activity [87] and to prevent HCV replication [16] NS5A-driven IRES-mediated viral genome translation [11,88,89]. Bachmetov et al. found Quercetin as an active chemical that inhibits NS3 protease activity, consequently lowering HCV production [16]. The schematic diagram in Figure 3 presented the mechanistic approach of Quercetin to work against the Hepatitis C virus.

Quercetin also affects eicosanoid production, protects LDL from oxidation, reduces platelet aggregation, and enhances cardiovascular smooth muscle relaxation [90]. Finally, Quercetin has been discovered to suppress the activity of the enzyme diacylglycerol acyltransferase (DGAT) [91,92], which is involved in the assembly step of the HCV life cycle [93]. Antiviral research is particularly interested in Quercetin’s antiviral mechanism since this multi-level (direct and indirect) approach against HCV could maintain selective pressure against viral relapse [94].

Quercetin modifies the HCV life cycle at several steps [95]: It (i) inhibits HCV genome replication; (ii) affects the morphogenesis of infectious particles, thus decreasing HCV-specific infectivity; (iii) affects the virion integrity when applied directly onto HCV particles; and (iv) hampers the localization of HCV core protein to LDs. Quercetin targets both viral and host factors. HCV uses the lipid metabolism of the host to complete its life cycle, from assembly to replication. Quercetin inhibits HCV-induced regulation of mRNA levels of many genes involved in lipid metabolism, secretion, and absorption and prevents the increase of DGAT protein activity. Diacylglycerol acyltransferase-1 (DGAT1), an enzyme that synthesizes triglycerides (TG) in the endoplasmic reticulum, interacts with HCV core protein and is implicated not only in the formation of new lipid droplets (LDs) but also in the production of infectious HCV [93]; as one of the Quercetin’s targets, DGAT1 is a key host factor for HCV infection. Using LDs as a platform for HCV assembly has been hypothesized, and Quercetin-induced reductions in DGAT activity could reduce the LD size by reducing the neutral lipid content and hence the LD membrane area available for HCV assembly. Quercetin blocks the subcellular localization of core proteins to LDs, implying that Quercetin has a significant impact on lipid metabolism. Quercetin changes its infectivity when introduced directly to HCV particles. This medicine affects the virion integrity and virulence and could be used as a coadjutant in prophylaxis after unintended exposure to HCV to slow down the viral infection. Moreover, the infectivity capacity of the newly produced viral particles was reduced by Quercetin treatment. Furthermore, Quercetin can reduce HCV infectivity through at least two different pathways [94]: (i) When applied to producing cells, it affects the morphogenesis of infectious particles, and (ii) when applied to virions, it affects their integrity.

### 7.3. Quercetin against Dengue Virus-2 (DENV-2)

The Dengue virus (DENV) is an enveloped, single positive-stranded RNA virus of the family Flaviviridae, genus Flavivirus [96]. DENV is a mosquito-borne virus, which is transmitted to humans by *Aedes aegypiti* and *Aedes albopictus* mosquitos [68]. There are four different DENV serotypes (DENV1, DENV2, DENV3, and DENV4), and it is possible to be infected four times [97].

Dengue virus (DENV) infection is becoming one of the worst mosquito-borne human pathogens and a major public health concern worldwide. About half of the world’s population is now at risk. Annually, 390 million dengue infections occur, with 96 million symptomatic and 20,000 fatal cases [98]. The virus causes dengue fever and, in severe cases, leads to lethal hemorrhagic fever/shock syndrome. Patients may deteriorate to severe dengue characterized by intense hemorrhage, plasma leakage, and, in some cases, shock, organ impairment, and death [17]. It is believed that recovery from an infection can provide lifelong immunity against that serotype. Secondary infection by other serotypes increases the risk of developing severe dengue. DENV-2 is the most dangerous dengue strain because it causes more dengue hemorrhagic fever epidemics than the other serotypes [99]. There are currently no antiviral treatments or vaccinations available to control DENV infection. However, adequate fluid management and regular monitoring are essential for Dengue treatment and for avoiding fatal complications [100]. Despite extensive studies, effective anti-dengue medications remain unattended [100]. Repurposed medications, such as chloroquine, prednisolone, balapiravir, celgosivir, and lovastatin, have been investigated in clinical studies; however, they are not effective in lowering the viral load and antigenemia or creating a positive effect in dengue patients [17,101,102]. Natural compounds are an excellent way to discover new drugs. Many studies showed Quercetin has anti-dengue potential activities [17].

### 7.4. Mechanism of Quercetin against DENV-2 Virus

Figure 4 has summarized the mechanism of Quercetin to work against DENV-2. The RNA genome of DENV codes three structural proteins, the capsid protein C, membrane protein M and the envelope protein E (capsid, prM, and E proteins), and seven non-structural proteins (NS1, NS2a, NS2b, NS3, NS4a, NS4b, and NS5) [101,102,103,104,105]. The E protein plays a vital role in the viral entry process [97]. Normal infection with DENV-2 was completely inhibited with Quercetin. Quercetin can exert antiviral effects on DENV through various mechanisms. Many in silico studies showed that Quercetin interacts with E and NS1 [104], NS3 [68], and NS5 [106] proteins and may disrupt a distinct process of the viral replication cycle. In addition, Quercetin acts directly on cells, altering innate response signaling pathways. As a result, treated cells have a greater ability to initiate defense mechanisms to prevent virus multiplication. Some studies showed that Quercetin induces the reactivation of type 1 Interferon-mediated (IFN-mediated) JAK-STAT (Janus kinases-JAKs, signal transducer and activator of transcription proteins-STATs) pathway [107], triggering the transcription of genes involved in the antiviral response. DENV blocks the JAK-STAT pathway in several ways: (1) targeting STAT2 protein for degradation in the proteasome [108]; (2) retaining phosphorylated-STAT1 in the cytoplasm of infected cells, this avoids nuclear translocation and transcription of antiviral genes [109]; (3) inducing overexpression of suppressors of cytokine signaling 1 and 3 [110,111,112] and tyrosine phosphatases that down-regulate the antiviral response. Nonetheless, there is evidence that Quercetin restores the innate response at least by modulating the expression of suppressor of cytokine signaling (SOCS) proteins [113] and SHP-2 tyrosine phosphatase [106]. Another study found that Quercetin has antiviral properties in cells that are the main targets of dengue infection and can block DENV-2 serotype [114].

A complex interaction between the virus and the host immune system is responsible for severe dengue, such as increased infection of immune cells due to non-neutralizing antibodies, cross-reactive autoantibodies, and T cells, as well as dysregulation of cytokine, complement, and coagulation processes [115,116]. According to a prior study, antibody-dependent enhancement (ADE) of infection occurs during secondary or tertiary infections. As a result, ADE increases cytokine secretion, which plays an important role in the pathogenesis of severe dengue [96,99]. Many studies showed that high levels of interferon-alpha (IFN-α), IL-8, IL-6, tumor necrosis factor-alpha (TNF-α), IL-10, and IFN-γ are associated with different stages of severe dengue disease [103,104,108,117]. A recent study showed that Quercetin modulates the production of TNF-α and IL-6 [114].

### 7.5. Quercetin against Ebola Virus (EBOV)

Ebola virus (EBOV), a member of the family Filoviridae, is an enveloped, single-stranded, negative-sense RNA virus [118]. The Ebola virus is a zoonotic pathogen. The genus Ebolavirus consists of six species, among those the species Zaire ebolavirus is the most virulent species, resulting in up to 90% mortality [18]. The Ebola virus is a highly contagious disease that causes hemorrhagic fever in humans and other mammals [119]. According to the World Health Organization (WHO), several outbreaks of the Ebola virus have resulted in more than 28,000 recorded cases and at least 11,000 deaths globally, and another outbreak is currently ongoing.

The virus of Ebola causes a life-threatening hemorrhagic fever, which starts with a high temperature and gastrointestinal problems, progresses to severe coagulation problems, and leads to multiorgan failure and death [120]. Due to the difficulty of early detection and the significant number of deaths, Ebola Virus Disease (EVD) is a global threat [121]. There are no FDA-approved medications and proven specific treatments to treat the Ebola virus infection [18]. The low fatality rate and the decline of this outbreak could be due to better supportive care, earlier case diagnosis, and admission [122]. However, great efforts have been made in the development of EBOV therapeutics [121].

### 7.6. Mechanism of Quercetin against EBOV Virus

The EBOV genome is a single-stranded RNA, approximately 19,000 nucleotides long. It encodes seven structural proteins: nucleoprotein (NP), polymerase cofactor (VP35), (VP40), GP, transcription activator (VP30), VP24, and RNA-dependent RNA polymerase (L) [119]. Several antiviral techniques have combated the distinct steps of the viral life cycle, interfering with the first phases of infection, such as a viral evasion of the interferon (IFN) system, which is the most attractive therapeutic approach against many types of viruses. Two viral proteins, VP35 and VP24 have been identified to decrease interferon responses [18]. Various investigations have shown that natural chemicals [123] can block VP35 IFN-inhibitory function, restoring the IFN production pathway.

Quercetin is a small chemical compound, that is already used as a nutritional supplement, has well-studied toxicity and pharmacokinetics [124], and metabolism in animal models, and is a promising anti-EBOV drug. A Quercetin derivative, 3-β-O-D-glucoside (Q3G), has been demonstrated to inhibit the early steps of EBOV entry [53]. Another study showed that Quercetin inhibits EBOV VP24, leading to a partial restoration of the Interferon-sensitive response element (ISRE) expression, ISG15 mRNA transcription, and phosphorylated STAT1 (P-STAT1) nuclear transport in the presence of the viral protein. Docking studies suggest the putative Quercetin binding mode at the interface between VP24 and karyopherin-protein (KPNα5). The effect of Quercetin on EBOV replication showed that it has the potential to suppress viral replication in Human embryonic kidney 293 cells (HEK293T) and affects specifically EBOV evasion of the IFN pathway. Quercetin is the first drug to inhibit the IFN-inhibitory action of EBOV VP24, restoring the IFN signaling cascade and preventing viral infection [18]. The mechanism of Quercetin against the Ebola virus is schematically shown in Figure 5.

### 7.7. Quercetin against Influenza A Virus

Influenza A viruses are enveloped negative-sense, single-stranded, segmented RNA viruses that belong to the virus family Orthomyxoviridae [125]. Several antigenic drifts and shifts in HA and NA create different subtypes of this virus, of which three HA (H1, H2, and H3) and two NA (N1 and N2) have handled human epidemics and pandemics for centuries [125]. Annually, influenza affects approximately 5–10% of adults and 20–30% of children [126]. Worldwide, it could cause about 290,000–650,000 deaths in one season. Besides mortality, influenza can induce respiratory, diabetic, cardiovascular, renal, and neurological problems [126]. Lately, the swine influenza virus (H1N1) is spreading among humans, which is the most alarming because of its pandemic potential [127]. However, because of the influenza virus’s genetic shift and drift mechanism, vaccinations or medications employed as a main prophylactic against influenza viruses are ineffective in eradicating the entire influenza viral infection. As a result, it is difficult to control influenza pandemics around the world [128]. Oseltamivir and zanamivir are commonly used drugs against the inhibition of influenza virus neuraminidase and infection; however, it was found that some isolates of H1N1 were resistant to the oseltamivir [128]. There is an urgent need to produce a safer and more affordable drug for the total eradication of influenza virus infection in humans.

### 7.8. Mechanism of Quercetin against Influenza A Virus

The viral genome of the influenza virus codes several functional, structural, and nonstructural proteins, including HA, NA, PB1, PB2, PA, NP, M (1,2), and NS (1,2), all of which play significant roles in the viral life cycle, such as attachment and entrance, viral RNA and protein synthesis, packaging, budding and release [125] (Figure 6). HA (hemagglutinin) and NA (neuraminidase) proteins are usually the targets for antiviral drugs [129]. Quercetin and its derivatives mainly inhibit anti-influenza virus infection and replication through different targets, including viral M2 protein, messenger RNA (mRNA), and glycoproteins (HA and NA) [127,128].

#### 7.8.1. Inhibiting Influenza Virus Entry via Blocking of HA and NA

The first stage of the viral replication cycle is virus entry; hence, the first line of defense against viral infectivity is the prevention of viral entry, which is the most attractive antiviral strategy. The envelope protein HA plays an important role in viral entry [127] and is comprised of two subunits: HA1 subunit, which is constantly evolving in an unpredictable fashion; the other subunit, HA2, however, is highly conserved but largely shielded by the HA head domain [130]. Quercetin prevents influenza virus entrance by inhibiting the HA protein of the virus. According to mechanistic investigations, Quercetin has an active interaction with the HA2 subunit. The pseudo-virus-based drug screening system showed that Quercetin prevents the entry of the H5N1 virus [131].

Most of the antiviral drugs against the influenza A virus mainly target the viral surface protein NA for inhibiting viral infection, which is involved in the release of progeny virions from infected cells [127]. Many studies have shown that Quercetin produced from plants inhibits influenza virus infection via the NA inhibition pathway [13,127]. In vitro and in vivo experiments confirmed that Quercetin from a Chinese traditional plant has a greater binding affinity to the active NA sites of A/PR/8/34 (H1N1) [132]. Computational studies showed that the chemical structure of Quercetin was able to suppress the NA crystal structure in silico, proving that Quercetin treatment reduced influenza virus-induced cytopathic effect [132].

#### 7.8.2. Inhibiting Influenza Virus via Blocking of Viral RNA Polymerase

Modern antiviral research proposes that influenza viral polymerase act as a target for anti-influenza drug development. Since there is no significant structural and genetic change across different influenza virus types and strains [133], viral RNA polymerase consists of PA, PB1, and PB2 subunits, which play a significant role in viral RNA synthesis in the influenza virus. Using the well-known “cap-snatching” mechanism, this viral polymerase uses host pre-mRNA as a primer for viral mRNA transcription [133]. Quercetin inhibits RNA virus infections by blocking viral polymerase [133]. In addition, in vivo studies showed Quercetin 3-rhamnoside had a higher inhibitory effect on influenza virus mRNA synthesis [29]. Molecular docking studies showed Quercetin 7-glucoside acts as a blocker of influenza H1N1 virus polymerase via occupying the binding site of 7-methylated GTP on PB2 subunit by RNA polymerase inhibition assay [24]. However, the status of Quercetin aglycone as an influenza viral polymerase blocker has yet to be verified.

## 8. Quercetin in Preventing Viral Infection through Immunomodulation

Quercetin functions as a powerful immunomodulatory molecule due to its direct modulatory actions on several immune cells, cytokines, and other immune chemicals, and indirect actions modulated through anti-inflammatory and antioxidant modes [9,134] (Figure 7). Quercetin has been highly recommended as an immunomodulatory antiviral drug in various studies due to its capacity to prevent the first phases of viral infection as an immunosuppressor or immunostimulator. It interacts with proteases required for viral replication and reduces infection-related inflammation. One possibility for reducing the occurrence of infections could be to improve people’s antiviral immune response through a nutritious diet that includes pure Quercetin taken from natural extracts. There are several mechanisms and studies focusing on the immunomodulatory actions of Quercetin on the major human viruses, such as HCV, HSV-1, H1N1, HBV, SARS-CoV2, and HIV-1, summarized in Table 2.

During the influenza course, Quercetin was found to affect the state of cytokine production. Quercetin is known to possess mast cell stabilizing, modulating, and regulatory action on immunity and inflammation [137]. Additionally, Quercetin has an immunosuppressive effect on dendritic cell function [138]. It limits LPS-induced inflammation via inhibition of Src- and Syk-mediated phosphatidylinositol-3-Kinase (PI3K)-(p85) tyrosine phosphorylation and subsequent Toll-Like Receptor 4 (TLR4)/MyD88/PI3K complex formation that limits activation of downstream signaling pathways in RAW 264.7 cells [139]. It can also inhibit the FcεRI-mediated release of pro-inflammatory cytokines, tryptase, and histamine from human umbilical cord blood-derived cultured mast cells (hCBMCs). The fatal consequence of influenza is eminently found to be associated with a massive viral load along with high cytokine storm or hypercytokinemia, which recruits a variety of innate immune cells [125]. TNF-α and IL-27 were tested from two categories of pro-inflammatory and anti-inflammatory cytokines, respectively. One of the cytokines that was altered was IL-27, which can boost the production of IL-10 by antiviral CD4+ cytotoxic T lymphocytes (CTLs), which can effectively modulate excessive immune response injuries [140]. Quercetin may boost IL-27 synthesis while decreasing TNF production.

Nair et al. (2009) found that Quercetin significantly downregulated p24 antigen production, LTR gene expression, and viral infectivity in a dose-dependent manner in Normal Peripheral Blood Mononuclear Cells [141]. Quercetin was reported to significantly downregulate the expression of the pro-inflammatory cytokine, TNF-α with concomitant upregulation of anti-inflammatory cytokine IL-13 as measured by the gene expression and protein production. They concluded that significant downregulation of TNF-α by Quercetin could probably be attributed to increased levels of IL-13. A higher level of IL-13 is known to inhibit TNF-α production and HIV-1 infection [141]. These findings suggest that in addition to the downregulation of HIV entry co-receptors by Quercetin, differential modulation of pro-and anti-inflammatory cytokines expression could be the potential mechanism for the anti-HIV activity of Quercetin.

The antioxidant, anti-allergy, and immunostimulatory effects of Quercetin are achieved by its capacity to inhibit histamine release and thereby antiviral effects by reducing the release of proinflammatory cytokines [142]. As we said earlier, Quercetin can inhibit the secretion of TNFα, which avoids triggering NF-κB [143] pathways and subsequently disrupts the production of IL1β, TNFα, and IL6. Furthermore, Quercetin greatly lowers the production of CCL-2, a key chemokine that governs monocyte and macrophage movement and infiltration throughout the inflammatory process [144]. Quercetin suppresses CXCL8, which induces neutrophil chemoattraction in a concentration-dependent manner, as well as RELA activity and recruitment [145]. All mechanisms mentioned contribute to the anti-inflammatory and immunomodulating properties of Quercetin, and the targets mentioned above are important for the infection of SARS-CoV-2 and DENV, especially DENV because the level of IL-1β, TNF, CCL2, CXCL8, and IL-6 would be upregulated after Dengue infection.

Quercetin was identified as the first identified inhibitor of the Ebola virus protein (EBOV VP24) anti-interferon (anti-IFN) function. High EBOV virulence and its potential to suppress the type I interferon (IFN-I) system is a promising novel anti-EBOV therapy approach that identifies the molecules targeting viral protein VP24 which is one of the main virulence determinants blocking the IFN response. Hence, Fanunza et al. (2020) in their experiment showed that Quercetin was able to suppress the VP24 effect on IFN-I signaling inhibition [18]. The mechanism of action lies with the Quercetin’s significant restoring capacity of IFN-I signaling cascade, blocked by VP24, by directly interfering with the VP24 binding to karyopherin-α and thus restoring Signal transducer and activator of transcription 1 (P-STAT1) nuclear transport and IFN gene transcription [146].

Quercetin has been found to suppress lipopolysaccharide (LPS)-induced tumor necrosis factor α (TNF-α) production in macrophages and LPS-induced IL-8 production in lung A549 cells [147]. It has been found to inhibit LPS-induced mRNA levels of TNF-α and interleukin (IL-)1α glial cells. Inflammatory cyclooxygenase (COX) and lipoxygenase (LOX) are also reported to be suppressed by Quercetin [148]. Interestingly, recent improvements in the pathophysiologic understanding of COVID-19 revealed that the severity of COVID-19 is strongly associated with cytokine release syndrome (CRS), which is characterized by elevated tumor necrosis factor (TNF-α), interleukin (IL)-6, IL-2, IL-7, and IL-10. TNF-α is often upregulated in acute lung injury, trigger CR, S, and facilitates SARS-CoV-2 interaction with angiotensin-converting enzyme 2 (ACE2). As a result, TNF-α inhibitors are thought to be a useful therapeutic strategy for slowing illness progression in severe SARS-CoV-2 infection [129,149].

All these studies show how Quercetin and its derivates impact immunomodulatory effects and eventually show a wide spectrum of antiviral activities, and a better understanding of Quercetin’s mechanistic properties could help in the rational design of more potent flavonol-type drugs to defend the emerging and reemerging viral infections.

## 9. Research Insights and Future Use of Quercetin

The molecular mechanisms behind Quercetin’s antiviral effects are the suppression of viral enzyme activity such as neuraminidases, DNA/RNA polymerases, and proteases as well as the immunomodulatory TNF-α (a distinguishing mechanistic diagram presented in Figure 8). As a result, its efficacy in inhibiting certain viral enzymes could be linked to increased immune response, making it a promising antiviral treatment option. Because of the lack of proofreading activity of most viral polymerases, the viral genome regularly mutates. These mutations could hamper the efficacy of antiviral synthetic drugs. By addressing the many signaling pathways, other than those we mentioned, involved in viral infections, the combination of synthetic antiviral medicines and Quercetin would improve therapeutic techniques [6] while dietary use of Quercetin should equally be focused to get the utmost benefit from plant-based natural sources enriched with Quercetin or its derivatives. The dietary consumption of total flavonoids has been estimated to be more than 200 mg/day, whereas the intake of flavonols is about 20 mg/day, with Quercetin accounting for more than 50%, with a daily intake of about 10 mg/day [150]. A Japanese investigation supported these figures since the daily consumption of Quercetin was determined to be 16 mg/day [151]. More advanced research is still demanded to claim the specific-mechanisms oriented development of antiviral drugs or formulation of antiviral supplements of Quercetin or its derivatives to affirm the utmost use of it.

Regarding the importance and efficacy of Quercetin against various virus infections, it is gaining attraction among researchers as a potential therapy for recent SARS-CoV-2 (Family: Coronavirus) that has caused a global pandemic situation. There has been little research into the effectiveness of Quercetin on COVID-19. According to one study, Quercetin is the most effective chemical for binding to the virus’s Spike Protein (S) receptor, as well as it binds to the COVID-19 major protease active site more strongly than hydroxyl chloroquine [122]. Climate change, epigenetic factors, abuse of drugs and so many other issues are provoking microorganisms, especially viruses and bacteria to cause multiple types of diseases while viral infections are pondered to be emerging and remerging issues. Therefore, the mechanistic insights of this review will no doubt make a way to suggest the protection of newer viral infections and explore an effective and safer antiviral drug. Nonetheless, Quercetin-and its derivatives-rich food supplements may also be helpful against severe to acute viral infections.

## 10. Conclusions

Quercetin is mechanically evident to have a wider range of antiviral properties. Unmet therapeutic demand for several viral infections needs to be investigated to affirm the therapeutic benefit of Quercetin. This is because, compared to synthetic medicines, Quercetin exhibits antiviral action without affecting cell viability at higher concentrations and has no negative effects on patients. Several in vivo, in vitro, and in silico investigations also confirmed Quercetin’s effective antiviral properties, which were like those of commercially available antiviral medications. Clinical trials should be included in future studies, so Quercetin and its conjugated products can be used safely. This review includes the most recent developments of Quercetin against some viral diseases, which may highlight the mechanisms of Quercetin on a molecular level. Understanding the mechanisms behind the antiviral effects of Quercetin could lead to new insights into filovirus entry pathways and potential targets for viral treatments. The exact mechanism of Quercetin in virus-host interactions with other viral infections is worthy of future investigation.

## Figures and Tables

**Figure 1 molecules-28-00938-f001:**
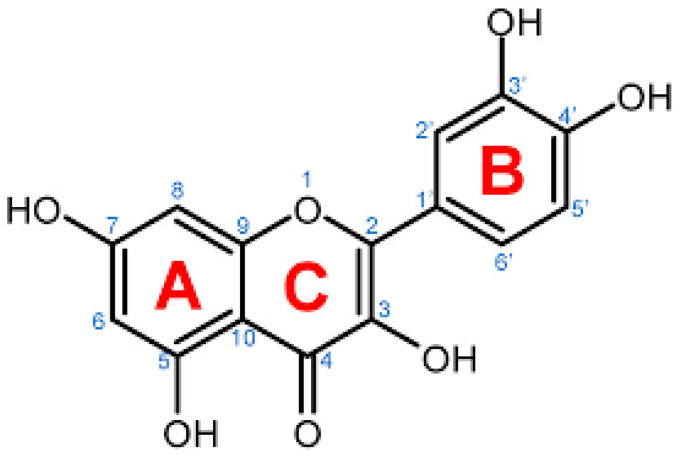
Chemical illustration of Quercetin taken from pubChem.

**Figure 2 molecules-28-00938-f002:**
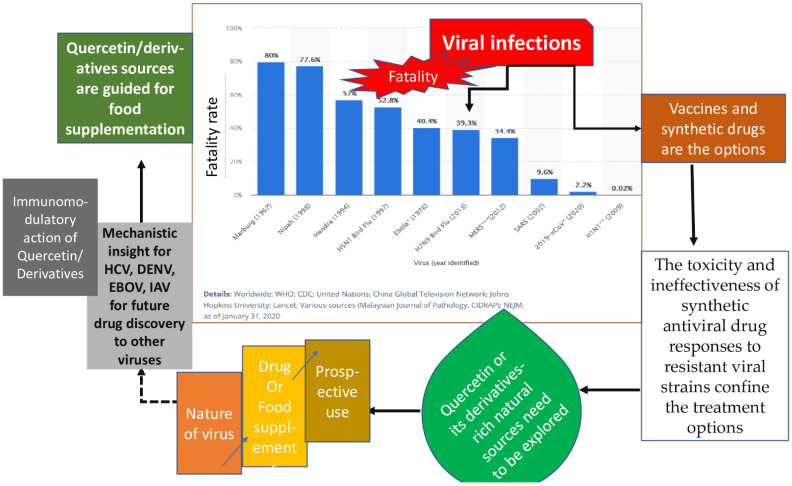
Schematic design aiming the approach and objective of this review.

**Figure 3 molecules-28-00938-f003:**
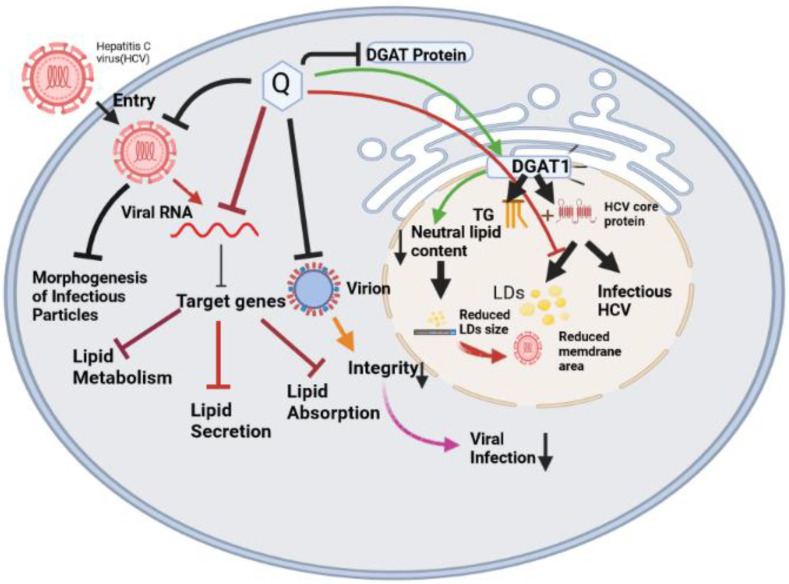
Effects of Quercetin in reducing the membrane area of Hepatitis C virus. The Hepatitis C virus entered the cellular system and is finally weakened through the protein molecule Diglyceride acyltransferase (DGAT, subtype DGAT1), which is inhibited by the administration of Quercetin. Inhibited DGAT downregulates the lipid synthesis, HCV core protein is eventually neutralized through the reduction of lipid droplets (LDs) size, which decreases the membrane size for HCV. HCV infections become attenuated.

**Figure 4 molecules-28-00938-f004:**
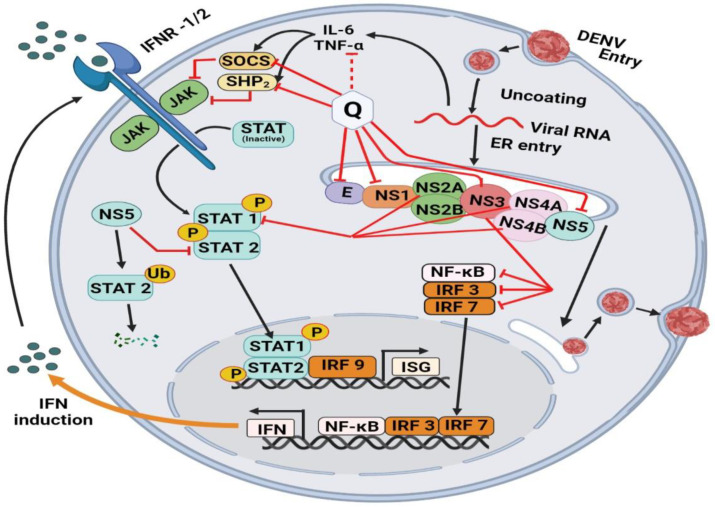
Effects of Quercetin in redirecting the virulence-mechanism of dengue virus. Ejection of Dengue virus particles from the cellular system with the involvement of cascade molecules, such as JAK, KFKB, STAT 1, STAT 2, NS2A, NS2B, SOCS, and SHP. The mechanism is assisted by the direct effect of Quercetin on the inhibition of the NF-kB pathway, which induces the production of a number of inflammatory and proinflammatory molecules. Interferon regulatory factors IFN, IRF3, and IRF7 are concomitantly reduced. The indirect effect of Quercetin is mediated by the inhibitory cascade of tumor necrosis factor, which affects the synthesis of IL6 and eventually leads to the restoration of Janus kinase (JAK)-signal transducer and activator of the transcription (STAT) pathway.

**Figure 5 molecules-28-00938-f005:**
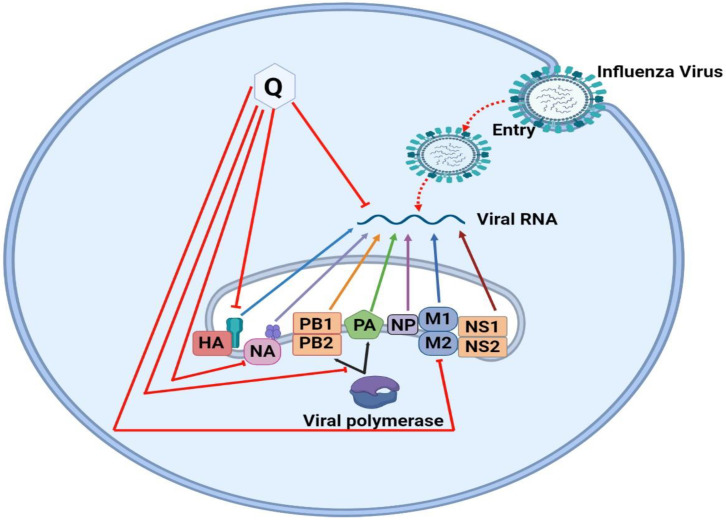
Effects of Quercetin on the Influenza virus. Quercetin augments in destructing the viral polymerase while its major target is viral RNA. Quercetin targets hemagglutinin (HA), which is a major determinant in subtype specificity. Subsequently the packaging subunit of viral RNA polymerase (PB1, PB2), acidic RNA polymerase (PA), nucleoprotein (NP), matrix protein (M1 and M2) and nonstructural protein (NS1 and NS2) subtypes of viral polymerase finally attenuate the RNA polymerase of Influenza virus through the removal of PB2, PA, NP and M segments.

**Figure 6 molecules-28-00938-f006:**
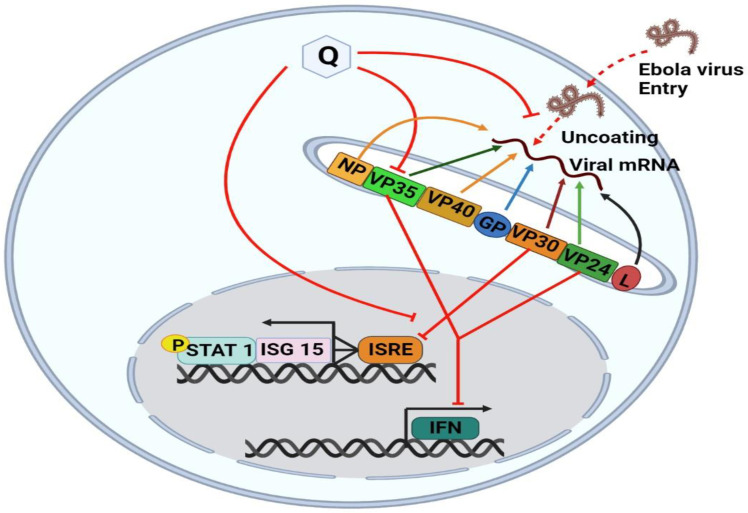
The unenveloped genetic material of the Ebola virus has interacted with Quercetin and with the aid of interferon-sensitive response element (ISRE) expression, ISG15 mRNA transcription, and phosphorylated STAT1 (P-STAT1) nuclear transport, replication of the Ebola virus approach to be halted. EBOV nucleocapsid proteins—nucleoprotein (NP), viral protein (VP35, VP24, VP30, VP40), and glycoprotein (GP) while NP, VP2,4 and VP 35 are necessary and sufficient to form transport-competent nucleocapsid-like structures. Quercetin directly restores the VP24 which is blocked by the Ebola virus and indirectly by using ISRE, ISG15 and STAT1.

**Figure 7 molecules-28-00938-f007:**
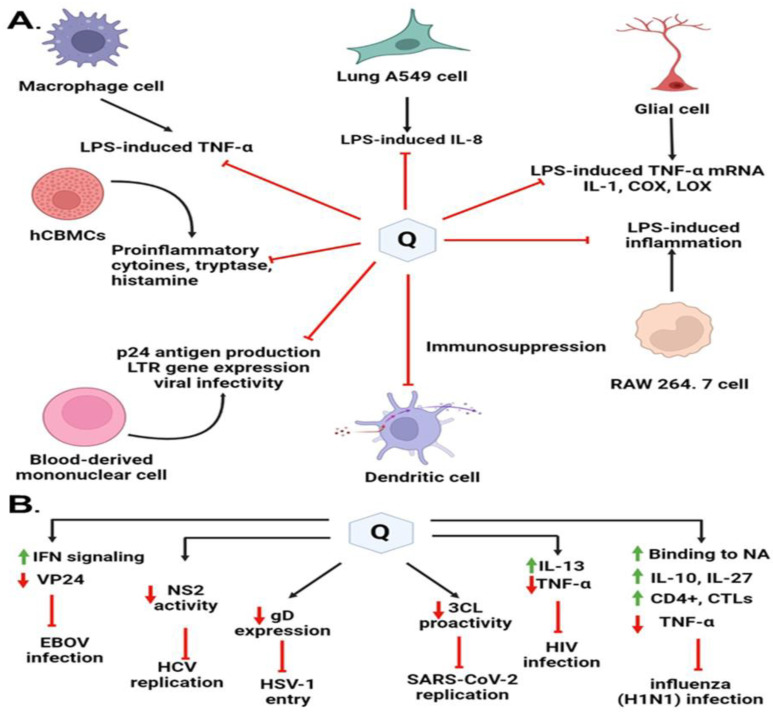
Immunomodulatory highlights on different antiviral mechanisms of action of Quercetin and its derivates. (**A**) Quercetin blocks virus entry or virus replication through interaction with viral proteins. (**B**) Immunomodulatory factors interleukins, tumor necrosis factor, and nonstructural viral proteins regulated by Quercetin play a key role in protecting the viral infections.

**Figure 8 molecules-28-00938-f008:**
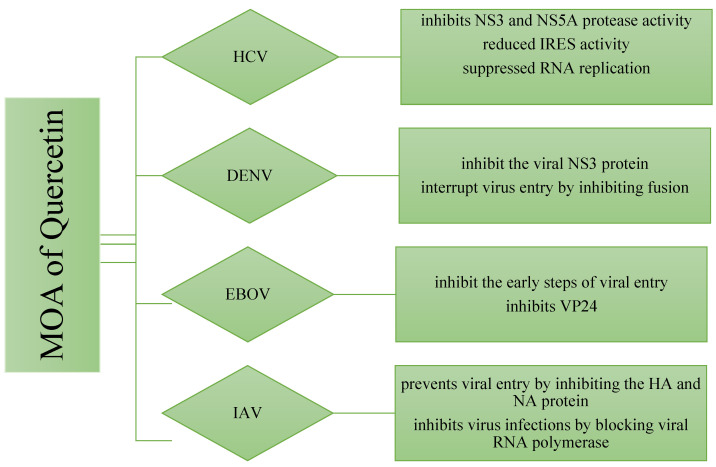
Differences in the mechanism of action of Quercetin for inhibiting four different types of viruses. The viral strains are denoted as hepatitis C virus (HCV), Dengue virus (DENV), Ebola virus (EBOV), and Influenzae virus (IAV).

**Table 2 molecules-28-00938-t002:** Different mechanisms of Quercetin to function as immunomodulatory agent.

Quercetin/Derivatives	Mechanism	References
Quercetin derivates	High binding activity on the capbinding site of influenza virus RNA polymerase PB2.	[133]
Quercetin	AntiHBV activity, inhibiting the formation of HBsAg and HBeAg. According to molecular docking, Quercetin forms very stable complexes with HBV	[135]
Quercetin	Latent HIV-1 gene expression is reactivated and nuclear factor κB nuclear translocation is induced	[39]
Quercetin	The docking studies show that Quercetin is a powerful inhibitor of the HCV NS2 protease.	[136]
Quercetin	Inhibits HSV entrance and NFB activation.	[47]
Quercetin	Inhibits 3CLpro and PLpro, with docking binding energies of 6.25 and 4.62 kcal/mol, respectively.	[10]

## Data Availability

All data will be available upon request to the authors.

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
