# Peer review of "Quercetin: A Functional Food-Flavonoid Incredibly Attenuates Emerging and Re-Emerging Viral Infections through Immunomodulatory Actions"

_molecules, 2023, doi:10.3390/molecules28030938_

Round 1

Reviewer 1 Report

The manuscript entitled "Quercetin: A Functional Food-Flavonoid Incredibly Attenuates Emerging and Re-Emerging Viral Infections through ImmunoModulatory Actions " by Md. Atiar Rahman, Fauzia Mahanaz Shorobi, Fatema Yeasmin, Srabonti Saha, Muhammad Abid Hasan Chowdhury, Mayuna Srisuphanunt, and Kazi Helal Hossai. It will be improved if the followings are addressed.

- The introduction is a bit short and I expect more insights as well as designed figures to explain the current research background to the readers.

- Table 1, the authors listed quercetin and its derivatives as well as other compounds like naringenin; it is unclear the aim of this table and seems a bit deviated from the topic of this article.

- The sections are not well-divided, please put more effort to reorganize with more subsections when necessary.

- Given the similar structure of quercetin and its derivatives, would all these compounds exhibit antiviral activity? If so, what is the similarity and difference in mode of action?

- The figure legends should be more elaborated and define all the abbreviations, e.g. it is unclear if Q is Quercetin among the figures. It is also weird that the Q in each figure is shown in different shapes and colors.

- At the end, what take home message would the authors suggest the reader? How many quercetin should be consumed? And should we consume it in diet for preventing viral infection or just use it as antivirals when we got virus infection?

- It is unclear why sometimes the word Quercetin starts in capital "Q" while sometimes in lower case "q". Same like in vitro and in vivo, sometimes in regular style and sometimes italized, please be consistent.

- Many of the cited references within the sentences are not coherent in style.

- Typos and unfriendly mode of English usage can be found.

Reviewer 2 Report

The article is thoroughly covered the therapeutic potential of quercetin on different viral infections. However, I have a serious concern about this manuscript. The following concerns are as follows-

1.       Authors have copied the previous published literature even without changing a sentence. The plagiarism is a serious violation of publication ethics. The whole introduction is taken from the author’s previous published article (https://doi.org/10.1007/s40203-022-00132-2). It is advised to avoid these practices while writing the article. The whole manuscript has more 35% plagiarism, which is huge (not acceptable). This needs a complete rewrite whatever the authors have copied from another article.

2.       I did not find the significance and importance of this article, what literature gap authors have considered before writing this review. Please consider writing why your review article adding something to the literature.

3.       Most of the literature authors have covered here is already published, for example, section 6 ‘isolation of quercetin’. These techniques are not adding much significance.

4.        Please confirm the reference of figures if you have taken or drawn yourself.

5.       Authors should consider covering the literature of recent viral infections such as COVID-19, Monkeypox with respect to flavonoids.

6.       It is recommended that author should mention the full form of abbreviation at the place of that word.

7.       Table 2, it is directly copied from previous published article. Authors are requested to rewrite the whole table.

8.       Kindly rewrite the whole section 10.

Round 2

Reviewer 1 Report

The authors have addressed most of my concerns.

Author Response

Thank you so much dear reviewer for your positive opinion

Reviewer 2 Report

The authors have addressed previously raised concerns about the manuscript; however, I have some minor concerns before recommending it for publication.

1.      In the Introduction, the very first sentence is started with “because”, it is recommended not to start the first word of the first sentence with because, please rephrase.

2.      The incorporation of figure 1 is not in right position, kindly insert at the end of the introduction.

3.      From lines 113-119, there are three sentences, and each sentence has started with quercetin, authors are advised to look into the grammar of the paragraph.

4.      From lines 155-159, “therefore, in this……or food supplements” these lines are not clear to understand, kindly correct it.

5.      The whole introduction is in 5 paragraphs, it would be better if you can merge the section into 4.

6.      Also, authors are requested to add a chemical illustration of quercetin in the introduction.

7. Lines 113-119; and 179-185, repetitive sentences, at both places author discussed the same thing; kindly correct.

8.      Line 272, in both in vitro and in vivo or “both in vitro and in vivo”. Kindly correct.

9.      Page 9, lines 285-290; “Many central nervous…… reduces neurodegeneration”. The description has no relevance with viral infection.

10.   Page 16, lines 570-572, too general lines, and within the same manuscript, authors have written this lines so many times, kindly not to do so.  Correction required.

11.   Figure 7, should be incorporated in the relevant section at the right place.
